# Electrochemical Characterization of Two Gut Microbial Strains Cooperatively Promoting Multiple Sclerosis Pathogenesis

**DOI:** 10.3390/microorganisms12020257

**Published:** 2024-01-25

**Authors:** Divya Naradasu, Waheed Miran, Akihiro Okamoto

**Affiliations:** 1Oral Microbiology, Bristol Dental School, University of Bristol, Dorothy Hodgkin Building, Whitson Street, Bristol BS1 3NY, UK; divya.naradasu@bristol.ac.uk; 2International Center for Materials Nanoarchitectonics, National Institute for Materials Science, 1-1 Namiki, Tsukuba 305-0044, Ibaraki, Japan; waheed.miran@scme.nust.edu.pk; 3School of Chemical and Materials Engineering, National University of Sciences and Technology, Islamabad 44000, Pakistan; 4Research Center for Macromolecules and Biomaterials, National Institute for Materials Science, Tsukuba 305-0044, Ibaraki, Japan; 5Graduate School of Chemical Sciences and Engineering, Hokkaido University, North 13 West 8, Kita-ku, Sapporo 060-8628, Hokkaido, Japan; 6Graduate School of Science and Engineering, College of Science and Engineering, University of Tsukuba, 1-1-1 Tennodai, Tsukuba 305-8573, Ibaraki, Japan; 7Living Systems Materialogy (LiSM) Research Group, International Research Frontiers Initiative (IRFI), Tokyo Institute of Technology, 4259 Nagatsuta-cho, Midori-ku, Yokohama 226-8501, Kanagawa, Japan

**Keywords:** whole-cell electrochemistry, gut microbes, extracellular electron transport, multiple sclerosis

## Abstract

In this study, we explored the extracellular electron transfer (EET) capabilities of two bacterial strains, OTU0001 and OTU0002, which are demonstrated in biofilm formation in mouse gut and the induction of autoimmune diseases like multiple sclerosis. OTU0002 displayed significant electrogenic behaviour, producing microbial current on an indium tin-doped oxide electrode surface, particularly in the presence of glucose, with a current density of 60 nA/cm^2^. The presence of cell-surface redox substrate potentially mediating EET was revealed by the redox-based staining method and electrochemical voltammetry assay. However, medium swapping analyses and the addition of flavins, a model redox mediator, suggest that the current production is dominated by soluble endogenous redox substrates in OTU0002. Given redox substrates were detected at the cell surface, the secreted redox molecule may interact with the cellular surface of OTU0002. In contrast to OTU0002, OTU0001 did not exhibit notable electrochemical activity, lacking cell-surface redox molecules. Further, the mixture of the two strains did not increase the current production from OTU0001, suggesting that OTU0001 does not support the EET mechanism of OTU0002. The present work revealed the coexistence of EET and non-EET capable pathogens in multi-species biofilm.

## 1. Introduction

The field of microbiology has seen significant advancements in understanding the extracellular electron transfer (EET) mechanism, particularly in environmental bacteria. These bacteria transfer electrons to extracellular solids like Fe (III) and Mn (IV) during anaerobic respiration [1,2,3]. This process, crucial for their metabolic activities, involves either direct or indirect electron transfer mechanisms [4,5]. In the direct EET mechanisms, *c*-type cytochrome complexes, which are multiheme proteins, are integral in transporting electrons across the outer membrane of these bacteria. This has been particularly well-documented in strains like *Shewanella oneidensis* MR-1 and *Geobacter sulfurreducens* PCA. These outer membrane cytochromes (OMCs) play a pivotal role in a wide array of electroactive microbes [4,6,7,8]. Notably, a few micromolars of endogenous riboflavin (RF) and flavin mononucleotide (FMN) significantly boost the rate of direct EET via the OMCs as a non-covalently bound cofactor [9,10]. The indirect EET mechanism can be enhanced by both endogenous and exogenous redox mediators. These mediators facilitate electron shuttling between microbes and electrodes, governed by diffusion-limited kinetics [11,12]. Therefore, indirect EET mechanisms usually require concentrations of around 100 micromolars of the soluble redox mediator to achieve a similar level of current enhancement with the bound flavins [9,13]. The occurrence of these EET mechanisms has been recently uncovered in diverse and unique ecological niches, including those involving human-to-animal pathogens [14,15].

Certain well-known human pathogens exhibit electrogenic behaviour, suggesting a link between their physiology and electrochemical activities. For instance, *Listeria monocytogenes*, a human gut-associated pathogen, can produce an electric current, which correlates with the extent of biofilm formation [14,16]. Further, *Enterococcus faecalis* [17,18] and a couple of human gut-isolated strains that have high genetic similarity with *Klebsiella pneumoniae* and *Enterococcus avium* are capable of EET [19]. *Pseudomonas aeruginosa*, one of the key players in human infections, is known to secrete high concentrations of phenazines for indirect EET mechanism [20]. Each EET mechanism is suggested to be involved in microbial physiology, facilitating the bacterium’s growth and pathogenicity within the host environment.

Consistent with such ideas, pathogens from oral biofilm have the EET capability and potentially make a thermodynamically downhill electron transport pathway within a 100 µm polymicrobial biofilm structure [15,21]. These microorganisms, within polymicrobial aggregates, are known to engage in interspecies electron transfer (IET), in addition to hydrogen or formate transfer, to boost their metabolism under anaerobic conditions [22,23]. *Streptococcus mutans*, *Corynebacterium matruchotii*, *Porphyromonas gingivalis*, and *Aggregatibacter actinomycetemcomitans* produce electric current associated with the presence of the cell-surface redox agents. The energy diagrams for these redox agents indicate a redox potential cascade from the anaerobic core to the periphery of the biofilm. This suggests that an IET may play a key role in supporting microbial metabolism in the anaerobic regions of the biofilm [15]. Understanding the mechanisms of IET is therefore vital for managing the activities of complex polymicrobial biofilms. However, the presence of multiple species complicates this task, requiring experimental approaches beyond omics studies, such as metatranscriptomics, due to their intricate nature [24]. A recent study identified the biofilm formation of two gut microbial strains, phylogenetically similar to *Lactobacillus reuteri* (OTU0001) and *Allobaculum stercoricanis* DSM 13633 (OTU0002), cooperatively induce multiple sclerosis (MS), an autoimmune disorder [25]. While the EET-capability of these strains has not been studied, the two-strain-biofilm system could be a model system for studying the interspecies electron transfer in biofilm formation and its potential impact on host-microbe interaction.

The present study investigated the EET capability of two gut microbes (OTU0002 and OTU0001). Using techniques such as single potential amperometry and differential pulse voltammetry, we aim to reveal the specific EET activities of these microbes. Our experimental approach, as shown in the schematic of the overall study mechanism in Appendix A, involves investigating direct and mediated current EET through supernatant replacement in a three-electrode chemical reactor, providing a thorough analysis of the electrical characteristics associated with these microbial strains.

## 2. Materials and Methods

### 2.1. Cell Culture Preparation

Pure strains of operational taxonomic unit (OTU) 0001 and 0002, obtained from RIKEN, Japan, were pre-cultured in a sterilized LB medium aerobically. All cultures were incubated at 37 °C until they reached the late exponential phase of growth. The cells were then centrifuged at 7200 rpm and 37 °C for 10 min. The resulting cell pellet was washed twice with a defined medium (DM) to eliminate any cell-secreted metabolites from the previous culture medium. The cell cultures were handled in a COY anaerobic chamber filled with 100% nitrogen. The DM was prepared using previously described methods [26] with some modifications. DM contained (per Liter) the following components: NaHCO_3_, 2.5 g; CaCl_2_∙2H_2_O, 0.09 g; NH_4_Cl, 1.0 g; MgCl_2_∙6H_2_O, 0.2 g; NaCl, 10 g; HEPES, 7.2 g.

### 2.2. Transmission Electron Microscopy

To collect cells at the exponential growth phase, 2–4 mL of 20 mL pre-cultures were centrifuged at 6000 rpm for 10 min. The cells were then immediately fixed in solutions containing 2% paraformaldehyde and 2.5% glutaraldehyde on ice. All further manipulations were conducted in 2 mL Eppendorf tubes after fixation. The washing process was done in 50 mM Na^+^-HEPES (pH 7.4, 35 g/L NaCl) with 5 × 1.5-mL washes by gently resuspending the pellet and centrifugation (5000× *g*, 4 min). Sequential metal enzyme-reactive 3, 3′-diaminobenzidine −H_2_O_2_ staining, OsO_4_ staining, and resin embedding procedures were conducted by following the previously described method [23]. The obtained resin blocks were sectioned at 80 nm with a diamond knife (DiATOME, ultra 35°), and floating sections were mounted on copper microgrids (Nishan EM). Thin sections were examined and imaged using a JEM-1400 microscope operated at 80 kV.

### 2.3. Whole-Cell Electrochemical Analysis of Pure Strains

Electrochemical measurements were conducted in single-chamber, three-electrode reactors. Indium tin-doped oxide (ITO) grown on a glass substrate by spray pyrolysis deposition (SPD Laboratory, Inc., Hamamatsu, Japan) was used as the working electrode (resistance, 8 Ω/square; thickness, 1.1 mm; and surface area, 3.14 cm^2^) and was placed at the bottom of the reactor. Platinum wire and Ag/AgCl (sat. KCl) were used as counter and reference electrodes, respectively. DM containing 10 mM glucose (4.8 mL) was injected into the electrochemical cell as an electrolyte, and the solution was purged with N_2_ gas for at least 15 min to remove the dissolved oxygen. The electrochemical cell was then filled with 0.2 mL of fresh cell suspension of OTU0001 and OTU0002, prewashed with DM, and injected to a final optical density of 0.5 and 1 at 600 nm under a potentiostatic condition of +0.4 V vs. the standard hydrogen electrode (SHE). During the electrochemical measurements, the reactor temperature was maintained at 37 °C, the reactor was operated without agitation, and experiments were run for about 24 h. Single-potential amperometry and differential pulse voltammetry (DPV) were measured with an automatic polarization system (VMP3, BioLogic Company, Seyssinet-Pariset, France). DPV was measured under the following conditions: from −0.6 V to +0.7 V vs. SHE at a pulse height of 50 mV, pulse width of 0.3 s, step height of 5 mV s^−1^, and step time of 5 s [9].

### 2.4. Supernatant Exchange Experiments during the Current Producing Condition

The medium in the electrochemical cell was removed; the electrode-attached cells were rinsed with N_2_-sparged DM twice at each replacement; and the headspace was continuously sparged with N_2_ during each replacement to avoid the leakage of oxygen into the electrochemical cell. The cell was refilled with N_2_-sparged sterile medium (10 mM glucose) and supernatant containing either planktonic cells or filtered medium.

### 2.5. Scanning Electron Microscopy (SEM)

For the scanning electron microscopy, ITO electrodes were removed from the reactors after performing the electrochemical measurements for about 24 h and slightly washed with 0.1 M phosphate buffer to remove the cells suspended on the electrode surface. Microbial fixation on electrodes was carried out with 2.5% glutaraldehyde diluted in 0.1 M phosphate buffer for more than one hour in the dark at room temperature. This was followed by washing three times in 0.1 M phosphate buffer (pH 7.4) for 15 min each. These washed samples were then dehydrated in 25%, 50%, 75%, and 100% ethanol gradients for 15 min each. Ethanol gradient dehydrated samples were exchanged thrice with 100% t-butanol and finally freeze-dried under a vacuum. The dried samples were coated with platinum and then observed using a Keyence VE-9800 microscope.

## 3. Results and Discussion

### 3.1. Electrochemical Characterization of OTU0001 and OTU0002

To evaluate the electron transfer abilities of gut pathogens OTU0002 and OTU0001, we examined their EET capability by measuring current production (*i*_a_) via single-potential amperometry with the ITO working electrode poised at +0.4 V vs. SHE. Following the introduction of bacterial cells into the reactor, a significant rise in net current of about 60 nA cm^−2^ was noted at a final optical density of 0.5 at 600 nm (OD600 nm). This increase immediately occurred in the presence of 10 mM glucose (blue line, Figure 1A). In contrast, no significant current was produced in the presence of OTU0001 (red line, Figure 1A) and the absence of microbes (black line, Figure 1A). These findings clearly imply that OTU0002’s EET capacity in conjunction with glucose oxidation.

After stabilizing the current production, a differential pulse voltammogram was conducted to investigate the electrochemical properties of the cell suspension. The presence of OTU0002 resulted in a broad oxidative peak ranging from approximately −0.3 V to ~+0.1 V vs. SHE, with the maximum peak potential (*E^p^*) at around −0.13 V (SHE) (as shown in Figure 1B).The large half-peak width of more than 200 mV (vs. SHE) is consistent with multi-heme outer-membrane cytochromes but not with soluble electron mediators such as riboflavin with that of 60 mV (vs. SHE) [27]. DP voltammetry did not show any significant peak current in the absence of microbes or the presence of OTU0001 (as shown in Figure 1B, black and red line). After several washes and dehydration, cellular attachment was observed on the ITO electrode surface for both strains using scanning electron microscopy (Figure 1C,D). This implies that the voltammetric signal is a result of the presence of redox enzymes on the surface of the microbe of OTU0002.

The assignment of the broad redox signal was supported by the change of current production upon electrolyte exchange experiments (Appendix A). Even after supernatant transfer to a fresh medium, a similar peak current with a broad potential window was observed on DP voltammograms (Appendix A, black and red lines), indicating that exogenous or endogenous soluble redox-active compounds are not the primary components of the redox signal. Therefore, the redox substrates on the cell surface are the most probable candidates attributable to the electrochemical signal. While DPV of spent cell-free spent medium exchange resulted in a slightly shifted peak potential (−0.14 V) and a second peak at −0.26 V, distinct from the main peak at −0.14 V solely coming from the cells remained on the electrode surface after fresh medium exchange (Appendix A, red line). The electrochemical redox peak might be attributed to proteins that have numerous redox molecules with varying redox potentials, as the combinatorial overlap of multi-redox processes increases the apparent peak breadth, resulting in a broad peak appearance.

We also performed an in vitro analysis utilizing the redox-dependent DAB chemical staining technique to identify cell-surface redox chemicals [23]. The two bacterial strains, OTU0002 and OTU0001, were grown under the same conditions and stained with DAB, and cross-sections were imaged under a transmission electron microscope (TEM). When DAB was negative, the cross-sectional images showed that OTU0002 cells stained with DAB did not exhibit any black precipitate (Figure 2A). Nonetheless, there was a noticeable distinction with thicker cellular margins in the presence of H_2_O_2_ (DAB positive, Figure 2B) as opposed to the absence of H_2_O_2._ On the other hand, the DAB-negative and -positive images of OTU0001 exhibited minimal black precipitation on the stained membrane (Figure 2C,D). These data suggest the presence and absence of redox substrate on the cell surface in OTU0002 and 0001, respectively, consistent with electrochemical analysis.

It is expected that cell-surface redox proteins would be expressed given that OTU0002 showed evidence of current production capability (Figure 1A). Thus, the redox staining seen on the cellular membrane further proved that OTU0001 was not capable of transferring electrons or that it has a redox centre that is less reactive to H_2_O_2_ oxidation. Our findings suggest that OTU0002 has extracellular or outer membrane proteins with metal centres that can catalyse H_2_O_2_ oxidation since the DAB process involves reaction centres that contain transition metals [28]. Therefore, a comprehensive analysis of cell surface proteins is necessary to identify the membrane proteins in these disorders, and this will be a critical area of focus for the future of our studies. Although the electrochemical characterization was conducted without identifying EET genes, our data on DAB suggest that it may be worthwhile to utilize gene expression analysis and gene-deletion techniques in future studies. This is because no EET genes have been identified in OTU0002 in the current literature [14].

To examine the coupling of metabolism with microbial current production, we analysed the activity of cells directly attached to the electrode during EET with Nano-SIMS by measuring the uptake of ^15^N (the only nitrogen source) by individual cells that most likely contributed to the current production [23,29]. After electrochemical experiments, we rigorously washed the electrode surface to remove planktonic or weakly attached cells on the electrode to analyse the anabolic ^15^N assimilation of cells by measuring ^15^N/N_total_ (%) and subtracting the natural abundance of ^15^N. In the single-potential amperometry condition, OTU0002 under the current production condition presented a significantly higher ^15^N (0.4%) anabolic activity during EET (Figure 3A). In contrast, the ^15^N assimilation was almost tenfold smaller in the OCV condition (Figure 3B,C). This distinct anabolic metabolic activity demonstrates the coupling with the current production of OTU0002.

### 3.2. Potential Involvement of Exogenous and Endogenous Redox Mediators on the Microbial Current Production of OTU0002

To investigate the EET capability of OTU0002, we examined the effect of cell density on its current production. We found that at ODs of 0.1 and 0.2, there was no significant increase in current production (Figure 4A). However, when the cell density was at an OD_600_ of 0.5, the current production increased by approximately 4 times compared to the OD_600_ of 0.1. Moreover, even though the cell density was enough to cover the whole electrode surface at an OD_600_ of 0.5 (Figure 4A), the current production was higher (~173 nA cm^−2^) at an OD_600_ of 1.0, indicating a non-linear relationship with cell density (Appendix A). It is important to note that when the electrode surface is fully covered at OD_600_ 0.5, the microbial current production is not only from cells attached to the electrode surface but also from those without direct attachment.

To quantify the contribution of planktonic cells in current production, we swapped the medium with the cell-removed filtered supernatant collected from a reactor (Appendix A). The current production did not increase and stayed at a low level. Given planktonic cells were also removed from the reactor and some cells should be attached to the electrode as SEM represents, this result indicates the dominant contribution of planktonic cells on the current production by OTU0002. The OTU0002 transfers electrons either through accumulative redox mediators in the supernatant or intercellular electron transfer via direct electron exchange among the cell-surface redox reagents [30].

We investigated the impact of external redox-active additives, such as Riboflavin (RF) and Flavin mononucleotide (FMN), which are ubiquitous exogenous redox substrates in gut environments [31,32], on current production (Figure 4B). The addition of RF and FMN to the reactor increased current production over threefold compared to the absence of redox-active additives (Figure 4B). In addition to *E_p_* of −0.10 V vs. SHE assignable to the cell surface protein, sharp oxidative peaks assignable to free RF were observed with *E_p_* at −0.23 V (SHE) in the presence of RF in the DP voltammogram (Figure 4C). In the case of FMN, the cell-specific redox peak showed a shift towards positive potential at +0.1 V vs. SHE besides a sharp peak rise at ~−0.25 V vs. SHE specific to FMN. Therefore, these data suggest OTU0002 uses free FMN substrate as an electron shuttle (Figure 4D).

Given that OTU0001 does not have EET capability, we hypothesized that OTU0001 may produce redox active substrates to accelerate the metabolic activity of OTU0002. However, OTU0001 and 0002 did not show enhanced current production upon mixing two species in a single EC reactor (Appendix A). Contrary to the synergetic effect on multiple sclerosis by combinational OTU0001 and 2 in our electrochemical analysis, the mixture of equal cell densities of OTU0002 and OTU0001 in the EC reactor showed a decreased current production (~35 nA/cm^2^, Appendix A) compared to OTU0002 alone current production, implicating a non-synergetic dependency with each other and absence of Quorum sensing (QS) mediated communication, thus electron transfer reduced to half. However, these data clearly indicate the impact of OTU0001 on OTU0002 either by diminishing its activity or sharing the metabolic electrons. The decrease could be due to OTU0001’s antimicrobial activity as *L. reuteri*, which is similar, produces antimicrobial molecules that inhibit harmful microbes, and change the host’s microbiota composition [33].

Given, electron mediators affect interactions with the bacterial surface, eventually, the transfer of electrons can be enhanced to maintain redox homeostasis within cells. [34]. The electron transfer of OTU0002 associated with metabolism may work together to enhance pathogenicity and the development of biofilms. Considering the ability of cell-surface redox regents to facilitate lateral electron transport over microbial aggregation [35], the polymicrobial biofilm containing this electrogenic gut pathogen may serve as a synergistic vehicle for the progression of the disease’s symptoms through electron exchange. Furthermore, EET linked to population-level phenotypes can be crucial during microbial colonization and biofilm formation. This phenomenon allows microorganisms to coordinate their behaviour and adapt to environmental changes. It is also possible that the pure culture biofilm of the electrogenic pathogen has electrical conductivity. In environmental EET-capable bacteria, such as *Geobacter*, clonal biofilm has electric conductivity over distances on the centimetre scale [36] due to membrane-bound cytochromes or redox shuttlers in biofilms. Thus, OTU0002 may have the long-range electron transfer capability as proposed in Appendix A. Pathogen EET could be a potential target for the development of new antimicrobial strategies that disrupt the electron transfer pathway and prevent biofilm formation. Based on the non-electrochemical activity of OTU0001, which is a probiotic healthy commensal [33], combined with the absence of redox species in both DPV and TEM analyses, it supports the bias of EET capability toward pathogenicity.

### 3.3. Phylogenetic Relevance of OTU0002 with Other EET-Capable Pathogens

We analysed the 16S rRNA sequences of this gut pathogen to determine its phylogenetic relevance with previously reported gut pathogens capable of EET. Using the National Center for Biotechnology Information (NCBI) database, we performed sequence alignment and compared the ribosomal RNA gene sequences of OTU0002 with representative microbial community sequences from known EET-capable gut microbes such as *Listeria monocytogenes* [14], *Enterococcus faecalis* [17], and *Faecalibacterium prausnitzii* [37]. Additionally, we overlaid the electrochemically enriched EET-capable gut pathogens *Enterococcus avium* and *Klebsiella pneumoniae* [19] as illustrated in Figure 5. Interestingly, the analysis revealed that OTU0002 exhibited a close relationship with *Listeria*, a Gram-positive foodborne human gut pathogen. *Listeria* is known to possess an EET mechanism that is coupled with a soluble electron acceptor, flavin molecules along with a membrane-bound protein complex involved in its EET mechanism. Given flavin increased current production, the flavin-cofactor enzyme is a potential EET mechanism in OTU0002 because flavin-bound cell surface enzymes mediate EET in some model microbes [38]. While the physiological role of EET in *Listeria monocytogenes* has not been clarified, the mutant strains lacking the gene encoding EET-flavoenzymes are responsible for mouse model infection [14]. Upon the identification of the critical genes for EET in OTU0002, their impact on biofilm formation and multiple sclerosis, therefore, should be studied in mouse models.

## 4. Conclusions

Our analysis using electrochemistry reveals that OTU0002, a human pathogen that is directly linked to multiple sclerosis, has the ability to transfer electrons in a manner that is dependent on cell density and active metabolism. In contrast, OTU0001, which coexists with OTU0002 in the gut of mice with autoimmune diseases, did not produce significant electrical current under the same electrochemical conditions. It is possible that OTU0001 might benefit from receiving electrons from OTU0002 for IET within the biofilm leading to the progression of disease. The use of redox mediators such as flavins suggests that these pathogens might use the available redox shuttles in the gut niche to progress their EET. Studying the electron transfer mechanism and electron uptake capabilities of both strains could be a valuable line of future research. Such research would not only enhance our understanding of microbial electron transfer processes but also broaden our perspective on microbial interactions in diverse environmental and biological settings.

## Figures and Tables

**Figure 1 microorganisms-12-00257-f001:**
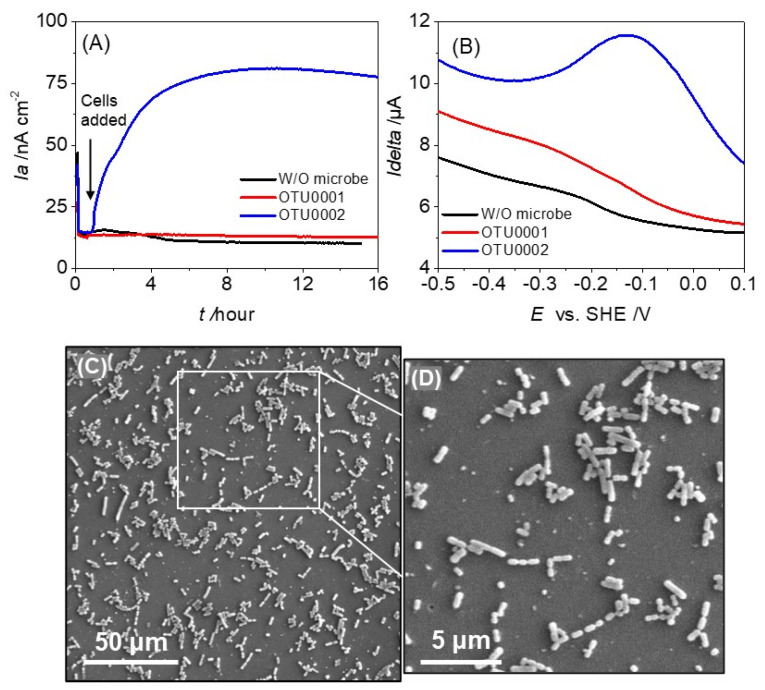
Evidence for extracellular electron transfer by the gut pathogen OTU0002. (**A**) Time vs. Current profile during the Single-potential amperometry (SA) over time in anaerobic reactors equipped with ITO electrodes at +0.4 V (SHE) in the presence of 10 mM glucose. OTU0002 showed a significant current production (blue line) compared to OTU0001 (red line) and without any microbes in the reactor (black line). (**B**). Differential pulse voltammograms in the presence of OTU0002 (blue line) and OTU0001 (red line). Data for sterile DM without any added microbes (black line) are also represented. (**C**,**D**) After 24 h of producing current with 10 mM glucose at +0.4 V (versus SHE), intact cells were observed attached to the electrode surface in scanning electron microscope images of OTU0002.

**Figure 2 microorganisms-12-00257-f002:**
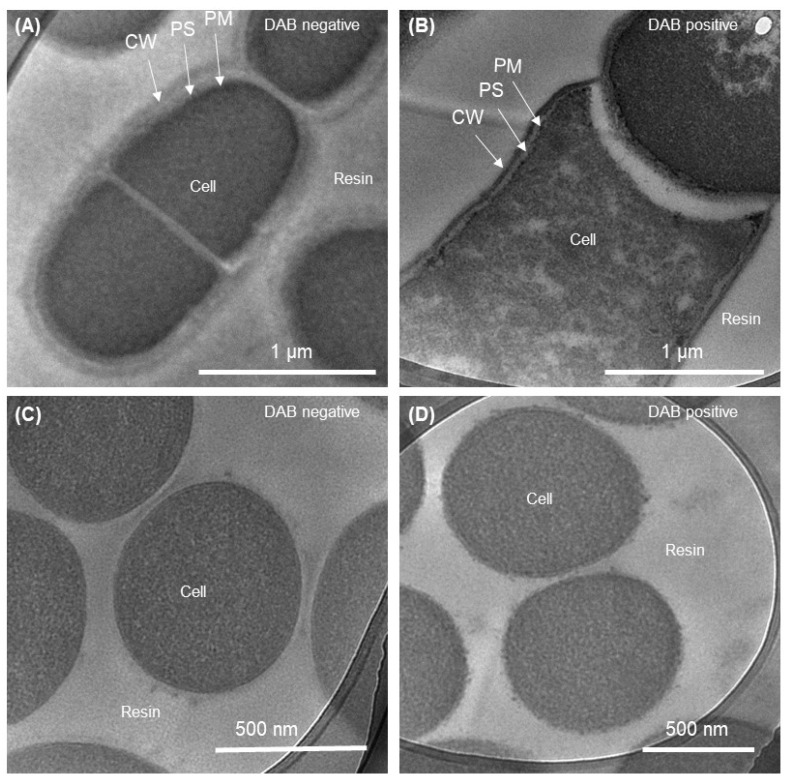
OTU0002 and OTU0001 cells stained with 3,3′-diaminobenzidene (DAB) are shown in a transmission electron microscopy (TEM) (**A**) and Positive DAB staining with the addition of H_2_O_2_ (**B**). Negative DAB stained OTU0001 in the absence and presence of H_2_O_2_ (**C**,**D**), respectively. Scale bars are shown in the images. CW: cell wall; PS: Periplasmic space; PM: Plasma membrane.

**Figure 3 microorganisms-12-00257-f003:**
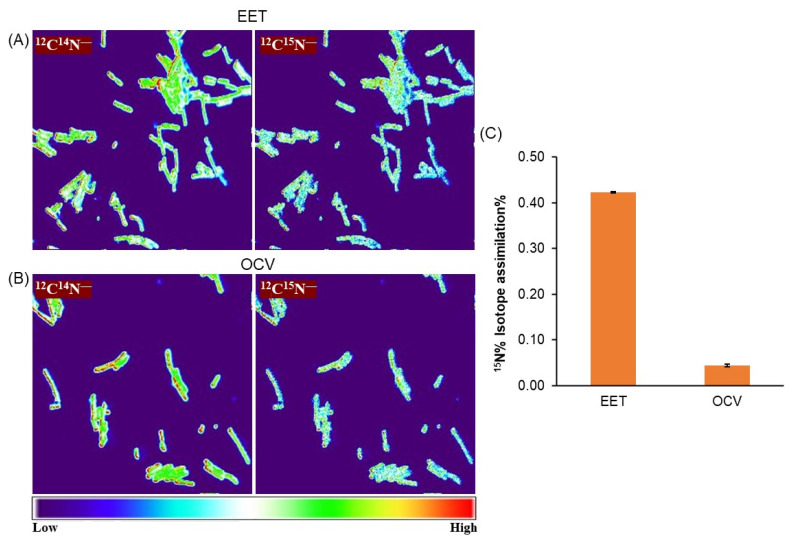
NanoSIMS images of OTU0002 cells attached to electrodes after EET (**A**) and OCV (**B**) measurements show the ^12^C^14^N^−^ and ^12^C^15^N^−^ ion pixel intensities. scale bar = 5 μm. The colour gradient bar indicates ion pixel intensity. Arrows indicate the cells. (**C**) Average assimilation with ± standard error mean (SEM) of ^15^N% assimilation of OTU0002 cells under EET and OCV, Number of cells selected for assimilation analysis, n = 64 and 61 for EET and OCV, respectively.

**Figure 4 microorganisms-12-00257-f004:**
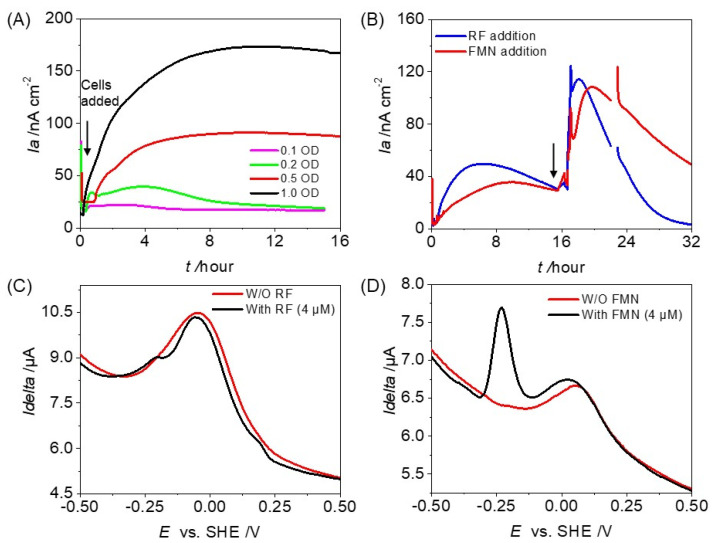
Population-induced EET capability and mediating electron transfer of OTU0002. (**A**) Current measurements in anaerobic reactors with ITO electrodes (3.14 cm^2^) at +0.2 V vs. Ag/AgCl in the presence of 10 mM glucose and OD_600_. 0.1, 0.2, 0.5, and 1.0 represent the initial cell densities added to the reactors. The arrow position indicates the time of cell addition in the electrochemical reactor. (**B**) Current production by OTU0002 in the absence (control) and the presence of external redox active additives, i.e., Riboflavin (RF) and Flavin mononucleotide (FMN). The arrow position indicates the time of mediator addition in the electrochemical reactor. (**C**,**D**) Differential pulse (DP) voltammogram of OTU0002 in the presence and absence (control) of redox-active additives (RF, FMN).

**Figure 5 microorganisms-12-00257-f005:**
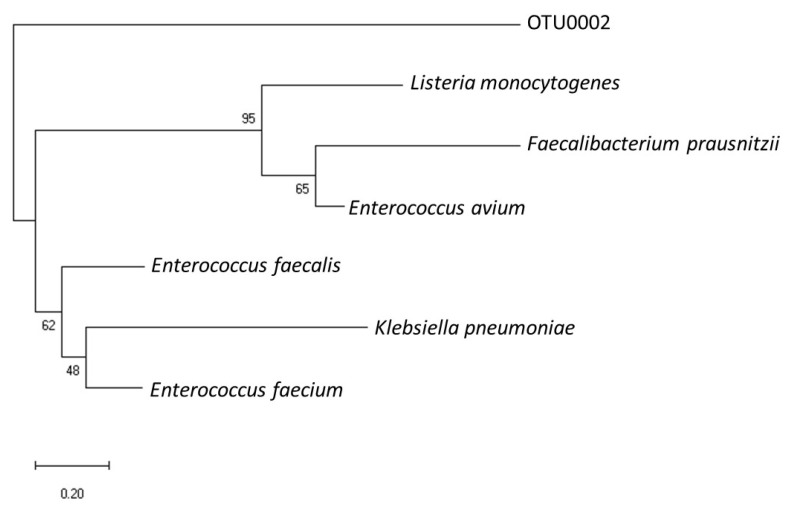
Phylogenetic tree depicting the identity of EET capable gut pathogens and the OTU0002, based on 16S rRNA sequence alignment, generated using MUSCLE; the neighbour-joining method was employed for phylogenetic tree construction (Scale bar: 0.20 substitutions per site).

## Data Availability

Data are contained within the article or Appendix A.

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
