# Peer review of "Electrochemical Characterization of Two Gut Microbial Strains Cooperatively Promoting Multiple Sclerosis Pathogenesis"

_microorganisms, 2024, doi:10.3390/microorganisms12020257_

Round 1

Reviewer 1 Report

Comments and Suggestions for Authors

The manuscript is interesting. The differences among registering-data are useful to bacterial characterization. 

Discussion could be enriched with details among suggested molecular interactions and effects triggering those registered data. Also, the details of suggestion about the role in Sclerosis should be clear and highlighted. Conclusions should be clear, concise and accurated related to that supported by observed data. 

Reviewer 2 Report

Comments and Suggestions for Authors

The current manuscript describes the research assessing the electrochemical characterization of two gut microbial strains OTU0001 and OTU0002 which are implicated in multiple sclerosis pathogenesis. The authors have performed various experiments to thoroughly understand yhe electrochemical characterization, redox mediator involvement, phylogenetic relevance of a strain with similar pathogens etc. The work has demonstrated the co-existence of extracellular electron transfer (EET) and non-EET capable pathogen in multi-species biofilm.
The manuscript in general is written very well and with precise descriptions. The experimental strategy is well organized and thoroughly executed as evident through the results. There are no major concerns. However, there are some minor questions which can be easily addressed by the authors following which the revised manuscript can be considered for acceptance. 

1.     Page 2, introduction, lines 78-80: it would be better to explain this in a few more sentences. What is the relevance of interspecies electron transfer mechanisms in biofilms? Why is it complex to study? This explanation will further provide more context on the relevance of the current study.

2.     Materials and methods in page 3, section 2.3, lines 130-132: what was the rationale behind choosing these specific conditions for measuring DPV?

3.     Page 5, lines 179-182 needs to be supported with literature reference.  

4.     Page 5, lines 192-195: the possible reasons for electrochemical redox peak and peak appearance are pointed out. However, there are no supporting literature references.

5.     Page 7, lines 251-253: how was it ascertained that the microbial current production is from both cells attached on the surface of electrode and those with no direct attachment?

6.     Page 10, conclusions, line 346: ‘Exploring the electron uptake abilities of these two strains could be a valuable line of inquiry’- please consider explaining this statement in detail. How can be this valuable considering its possible impact on disease and health? Since the context here is multiple sclerosis, the correlation specific to the disease needs to be explained here. This section should also highlight the limitations of current study. Future studies are briefly indicated in paragraph 2 of page 6, however it needs to be added to this section appropriately.

7.     Authors can also consider providing a schematic illustration (similar to current figure S4) as a separate figure in the beginning highlighting the overall mechanism of the study to improve the readability.

8.    Line 159 in section 3.1 of page 4, line 183 in page 5, line 224 in page 6- these needs grammatical corrections. Please consider re-reading the overall manuscript carefully for checking grammatical and spelling corrections.

Comments on the Quality of English Language

As indicated above, please re-read the manuscript and check for grammatical/ language/ spelling errors; rectify them as needed. A few instances have been pointed out in the review summary.

Reviewer 3 Report

Comments and Suggestions for Authors

File is attached 
